# Semi-Supervised Abdominal Organ and Pan-cancer Segmentation with Efficient nnU-Net

Ziran Chen[1,3], Taiyu Han[2,3], Xueqiang Zeng[2,3], Guangtao Huang[2,3], Huihui Yang[2,3], and Yan Kang[1,2,3]⋆

[1] College of Medicine and Biological Information Engineering, Northeastern University, Shenyang, China
[2] College of Applied Sciences, Shenzhen University, Shenzhen, China
[3] College of Health Science and Environmental Engineering, Shenzhen Technology University, Shenzhen, China
kangyan@sztu.edu.cn

**Abstract.** Abdominal organs serve as frequent sites for the manifestation of cancer, however, a prevailing gap exists in the availability of a widely accessible and precise segmentation model tailored to these organs and associated tumors. While nnU-Net has become a powerful baseline for medical image segmentation in recent years, its default configuration lacks the ability to leverage unlabeled data and falls short in terms of inference efficiency. To surmount these inherent constraints, we propose an improved approach based on nnU-Net. Our proposed method incorporates a semi-supervised algorithm that utilizes pseudo-labeling to effectively process unlabeled data within the nnU-Net framework. We improve the utilization of unlabeled data by generating high quality pseudo-labels with the default nnU-Net. Additionally, we reduce the network complexity of 3D U-Net and train a lightweight student model using a combination of labeled and pseudo-labeled data. In terms of performance, our lightweight student model achieved promising results on the validation set. The method yielded the average DSC of 0.8856 and NSD of 0.9451 in the process of segmenting 13 abdominal organs. For tumor segmentation, the average DSC and NSD were computed as 0.4258 and 0.3513, respectively. The average running time per case is 29s and the average GPU memory is 25411MB. In conclusion, our approach effectively addresses the limitations of nnU-Net, improving both inference efficiency and the utilization of unlabeled data. The encouraging results obtained in the FLARE 2023 challenge underscore the potential of our method to advance practical clinical applications in the field of medical image segmentation.

**Keywords:** Segmentation · Semi-supervised learning · Pseudo-labels.

## 1 Introduction

Accurate segmentation of abdominal organs is crucial for diagnosing and treating abdominal lesions [1], with CT imaging being widely used in clinical practice.

---

⋆ Corresponding author

Manual segmentation is time-consuming, labor-intensive, and prone to variability among observers [2]. AI development presents an opportunity to automate this process, reducing the burden on clinicians and enabling applications like surgical planning. However, deep learning models have become increasingly complex, requiring large amounts of labeled data [3]. Medical image segmentation, especially for intricate abdominal organs or diseases, requires pixel-level labels that can only be provided by experts. The growing size and resolution of images exacerbate these challenges. In clinical settings, there is a significant amount of semi-labeled or unlabeled data that cannot be effectively utilized with fully supervised learning approaches.

In this context, the practicality of semi-supervised segmentation methods has seen a notable increase, primarily attributed to their ability to effectively exploit small, accurately labeled datasets while leveraging larger pools of unlabeled data, resulting in enhanced model accuracy [4]. In recent years, the field of medical image segmentation has witnessed widespread application of semi-supervised learning techniques. These approaches exploit the global characteristics of data by incorporating unlabeled samples and can be broadly categorized into three types [5]: first, utilizing model predictions on unlabeled images to generate pseudo-labels for subsequent model training; second, jointly training the model with both labeled and unlabeled data; third, incorporating unlabeled images with prior knowledge (such as shape and location) along with labeled images during model training. Initially, pseudo-labeling was employed in early semi-supervised methods [6], and many existing frameworks have integrated this concept [7,8,9,10]. Presently, the latest advancements in semi-supervised segmentation revolve around consistency learning, adversarial learning, and entropy minimization. Furthermore, hybrid semi-supervised learning [11,12,13,14] has gained significant attention and application, where diverse methods are integrated to optimize the model and enhance segmentation performance.

In recent years, the nnU-Net framework [15] has been widely adopted for medical image segmentation. While it excels in fully supervised scenarios, it lacks built-in support for semi-supervised training. In practical clinical settings, time constraints for inference and limited labeled data availability hinder optimal efficiency with the default nnU-Net. The first-place solution of Flare 2022 challenge developed a semi-supervised learning framework based on nnU-Net [16], demonstrating the efficacy of pseudo-labeling methods in leveraging unlabeled data to enhance model robustness.

In this work, we propose a semi-supervised framework based on nnU-Net for abdominal organ segmentation, aiming to meet the requirements of fast inference and low computational cost while making the most of a limited amount of labeled data. We introduce high-quality pseudo-labels by utilizing a resource-intensive nnU-Net trained on fully labeled CT scan data to generate them for the semi-labeled data. In a lightweight nnU-Net, we jointly train the labeled images with pseudo-labels and the unlabeled images to obtain a model capable of fast inference. Additionally, we have implemented a method proposed in the literature [16,17] that utilizes efficient sliding window strategies based on prior

knowledge of abdominal organs to reduce the number of inference windows and leverages GPU for resizing data, leading to improved inference efficiency.

The principal contributions of our study can be outlined as follows:

- We propose a semi-supervised segmentation framework based on nnU-Net, which effectively utilizes unlabeled data to improve the segmentation accuracy of models.
- We performed compression optimization on the default nnU-Net. By reducing the input data size and using a narrower network width, we maintained high segmentation accuracy while minimizing GPU resource requirements, making it more suitable for practical applications and competition needs.

## 2  Method

To leverage unlabeled data, we initiate the process by training a teacher model using labeled data. We chose the default nnU-Net as the teacher model, which works well in most cases. Subsequently, we utilize the well-trained teacher model to generate predictions for the unlabeled data, thereby obtaining high-quality pseudo-labels that significantly enhance the training process. To optimize inference efficiency, we draw inspiration from the first-place solution of the previous Flare 2022 challenge [16] and downscale the default U-Net network of the nnU-Net framework to serve as a student model. The modified student model is trained by leveraging these pseudo-labeled data in combination with the labeled dataset. This approach ensures that the smaller student model achieves segmentation accuracy comparable to its larger teacher model while concurrently improving segmentation efficiency to meet the resource-constrained requirements. An overview of the framework we designed is shown in Fig. 1.

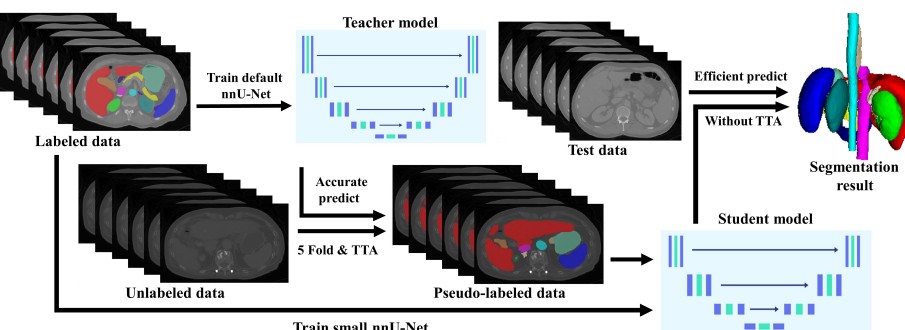

**Fig. 1.** Teacher-Student Semi-Supervised segmentation framework: During the model training, the teacher model infers on unlabeled data and utilizes high-quality pseudo-labeled data generated by the larger teacher model to train a more efficient student model.

### 2.1    Preprocessing

We divided our partially labeled dataset of 2,200 samples into two distinct training sets based on the labeled content. The first set consisted of 222 samples, each labeled with all 13 organs, and was used to train a teacher model for organ segmentation. The second set encompassed a collection of 597 samples, each labeled to encompass tumors and six additional organs, thereby serving as the foundation for training dedicated tumor segmentation models.This approach allowed us to train independent models for organ segmentation and tumor segmentation, enabling us to focus on the specific features and characteristics of each segmentation type, resulting in improved accuracy and efficiency of our analysis.

Regarding image preprocessing, we applied several steps to the FLARE 2023 dataset using the default nnU-Net settings. First, we cropped each CT scan to remove non-zero regions. Subsequently, the CT images were subject to clipping, specifically aligned with the 0.5 and 99.5 percentiles of the foreground voxels, followed by the implementation of Z-score normalization utilizing the global foreground mean and standard deviation.Finally, we resampled each scan to a specific uniform spacing to ensure consistency in the dataset. We used third-order spline interpolation and nearest-neighbor interpolation methods for the data and segmentation mask, respectively. This step was crucial since CT scan values are directly related to physical properties, and keeping them in a preprocessing state was essential for accurate analysis. The resampling spacing for the teacher model is [2.5, 0.86, 0.86], whereas for the student model, it is [4.0, 1.2, 1.2]. Overall, these preprocessing steps ensured that the FLARE 2023 dataset was suitable for meaningful and accurate analysis of abdominal tumors and organs.

### 2.2    Proposed Method

Our backbone network utilizes the 3D U-Net structure, which is commonly employed for training 3D medical images such as CT and MRI. However, it requires a significant amount of GPU memory. To enhance training speed and reduce resource consumption, a patch-based 3D U-Net approach can be adopted to lower network computing costs. The primary objective of 3D U-Net is to address the limitations of 2D U-Net when applied to anisotropic data. In the nnU-Net framework, the ReLU activation function is substituted with Leaky ReLU, batch normalization is replaced by instance normalization, and the network structure closely resembles the default 3D U-Net architecture.

In order to improve inference efficiency and computational cost, we made specific modifications to the 3D U-Net based on the default settings of the nnU-Net framework. Our small-scale 3D U-Net network takes input patches of size $32 \times 128 \times 192$, with a batch size of 2, and comprises four up-sampling and down-sampling layers. Each layer consists of 3D convolution, LReLU activation, and instance normalization. The initial layer of the 3D U-Net extracts 16 feature maps, while each downsampling process extracts a maximum of 256 feature maps. The structure of the 3D U-Net backbone network is depicted in Fig. 2.

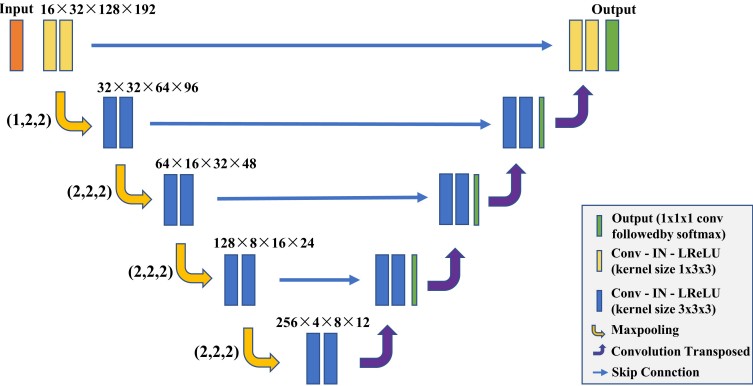

**Fig. 2.** The small 3D U-Net network architecture.

Loss function: We utilize the default composite loss function in nnU-Net, which combines the Dice loss and Cross-entropy loss. This composite loss function has been proven to exhibit robustness in various medical image segmentation tasks. [18].

### 2.3   Post-processing

Due to meet time constraints, we opted not to employ complex post-processing techniques. To expedite the inference process, Test Time Augmentation (TTA) was disabled, resulting in a 2x reduction in inference time. Given that we have two segmentation models, it is necessary to integrate the results of the organ segmentation and tumor segmentation models. The tumor segmentation results are superimposed on the organ segmentation results, and if there are any conflicts, the tumor segmentation results are given priority.

## 3   Experiments

### 3.1   Dataset and evaluation measures

The FLARE 2023 challenge is an extension of the FLARE 2021-2022 [19,20], aiming to aim to promote the development of foundation models in abdominal disease analysis. The segmentation targets cover 13 organs and various abdominal lesions. The training dataset is curated from more than 30 medical centers under the license permission, including TCIA [21], LiTS [22], MSD [23], KiTS [24,25], and AbdomenCT-1K [26]. The training set includes 4000 abdomen CT scans where 2200 CT scans with partial labels and 1800 CT scans without labels. The validation and testing sets include 100 and 400 CT scans, respectively, which cover various abdominal cancer types, such as liver cancer, kidney cancer, pancreas cancer, colon cancer, gastric cancer, and so on. The organ annotation process used ITK-SNAP [27], nnU-Net [15], and MedSAM [28].

The evaluation metrics encompass two accuracy measures—Dice Similarity Coefficient (DSC) and Normalized Surface Dice (NSD)—alongside two efficiency measures—running time and area under the GPU memory-time curve. These metrics collectively contribute to the ranking computation. Furthermore, the running time and GPU memory consumption are considered within tolerances of 15 seconds and 4 GB, respectively.

### 3.2   Implementation details

**Environment settings**  The development environments and requirements are presented in Table 1.

**Table 1.** Development environments and requirements.

| System | Ubuntu 18.04.4 LTS |
|---|---|
| CPU | Intel(R) Xeon(R) Gold 6146 CPU@3.20GHz |
| RAM | 256G |
| GPU | Two NVIDIA V100 16G |
| CUDA version | 10.2 |
| Programming language | Python 3.8 |
| Deep learning framework | torch 1.11, torchvision 0.12.0 |
| Specific dependencies | nnU-Net |

**Training protocols**  We employed nnU-Net's deep supervision loss, a methodology that incorporates the output layer into the loss calculation during each upsampling operation. To achieve this, different weights are assigned to the shallowest and deeper layers, with a weight of 1 for the shallowest layer and halved weights for each subsequent deeper layer. This combined approach integrates equally weighted dice and cross-entropy loss terms, facilitating comprehensive and effective training. To harness the benefits of ensemble learning, we trained five teacher models iteratively until reaching convergence. Subsequently, we utilized their collective predictions on the unlabeled samples to generate pseudo labels, which were then employed for training the student model. This teacher-student training paradigm with pseudo labels serves to leverage the information-rich unlabeled data effectively, enhancing the performance of the student model. For the specifics of our training procedure, including batch size, number of epochs, optimizer choice, and other relevant details, please refer to Table 2 and Table 3 for comprehensive information. These tables present a clear and organized overview of the training protocols, ensuring reproducibility and facilitating comparison with other methodologies in the field.

**Data Augmentation**  During the training of the teacher model, several data augmentation techniques were employed, including the use of Gaussian noise,

brightness adjustment, gamma correction, rotation, scaling, elastic deformation, and simulated low-resolution data. It is worth noting that for the training of the student model, only a subset of these techniques were utilized, specifically luminance adjustment, gamma correction, rotation, scaling, and elastic deformation.

**Table 2.** Training protocols for teacher model.

| Network initialization | "HE" normal initialization |
|---|---|
| Batch size | 2 |
| Patch size | Stage0 [64,192,192], Stage1 [48,192,192] |
| Total epochs | 1000 |
| Optimizer | SGD with nesterov momentum ($\mu = 0.99$) |
| Initial learning rate (lr) | 0.01 |
| Lr decay schedule | Poly learning rate policy:$(1 - epoch/1000)^{0.9}$ |
| Training time | 72 hours |
| Number of model parameters | 31.2M |
| Number of flops | 230G |

**Table 3.** Training protocols for student model.

| Network initialization | "HE" normal initialization |
|---|---|
| Batch size | 2 |
| Patch size | Stage0 [32,128,192], Stage1 [32,128,192] |
| Total epochs | 1500 |
| Optimizer | SGD with nesterov momentum ($\mu = 0.99$) |
| Initial learning rate (lr) | 0.01 |
| Lr decay schedule | Poly learning rate policy:$(1 - epoch/1500)^{0.9}$ |
| Training time | 24 hours |
| Number of model parameters | 6.1M |
| Number of flops | 140G |

## 4  Results and discussion

The DSC and NSD results for all experiments were obtained through the online validation leaderboard in the MICCAI FLARE 2023 challenge. Additionally, the detailed results of the 50 public validation sets were processed privately. The challenge organizers provided an efficiency analysis based on the Docker containers we submitted.

### 4.1   Quantitative results on validation set

Overall, the quantitative evaluation results on the provided validation and test sets are shown in Table 4. On the validation set, the average DSC for the 13 organs is 0.8856, with an average NSD of 0.9491. The DSC for tumor segmentation is 0.4258, with an NSD of 0.3513. On the test dataset, the average DSC for the 13 organs is 0.8856, with an average NSD of 0.9491. The DSC for tumor segmentation is 0.4258, with an NSD of 0.3513. Due to limitations in inference time and memory, our approach employed larger spacing for resampling the input data and used smaller patch sizes. This exacerbated the risk of losing contextual information when dealing with small targets. It can be observed that the segmentation performance for the right adrenal gland, left adrenal gland, and gallbladder, which are small-volume organs, is relatively poor.

**Table 4.** Quantitative evaluation results.

| Target | Public Validation | | Online Validation | | Testing | |
|---|---|---|---|---|---|---|
| | DSC(%) | NSD(%) | DSC(%) | NSD(%) | DSC(%) | NSD (%) |
| Liver | $97.37 \pm 0.55$ | $84.58 \pm 4.46$ | 97.30 | 98.76 | 95.91 | 97.15 |
| Right Kidney | $94.35 \pm 7.38$ | $93.98 \pm 8.70$ | 93.32 | 94.99 | 94.23 | 95.36 |
| Spleen | $96.46 \pm 1.65$ | $98.61 \pm 2.84$ | 95.67 | 97.96 | 95.90 | 97.96 |
| Pancreas | $85.70 \pm 6.25$ | $94.38 \pm 6.12$ | 84.53 | 95.77 | 88.58 | 97.10 |
| Aorta | $94.92 \pm 2.11$ | $99.49 \pm 1.56$ | 95.02 | 98.22 | 95.31 | 99.19 |
| Inferior vena cava | $93.61 \pm 2.13$ | $99.53 \pm 0.96$ | 93.21 | 96.45 | 93.42 | 96.96 |
| Right adrenal gland | $81.28 \pm 6.24$ | $99.95 \pm 0.19$ | 80.30 | 94.79 | 78.01 | 93.34 |
| Left adrenal gland | $78.52 \pm 7.14$ | $99.83 \pm 0.39$ | 78.08 | 92.59 | 77.41 | 92.50 |
| Gallbladder | $83.62 \pm 18.8$ | $93.93 \pm 19.9$ | 84.67 | 83.62 | 81.68 | 83.07 |
| Esophagus | $81.21 \pm 15.2$ | $96.85 \pm 14.4$ | 81.71 | 93.85 | 87.47 | 98.29 |
| Stomach | $92.11 \pm 3.63$ | $98.33 \pm 3.69$ | 92.63 | 97.34 | 92.33 | 97.43 |
| Duodenum | $81.67 \pm 7.49$ | $96.85 \pm 3.76$ | 82.27 | 95.18 | 85.27 | 96.97 |
| Left kidney | $92.26 \pm 12.5$ | $97.98 \pm 7.72$ | 92.53 | 94.28 | 92.99 | 94.69 |
| Tumor | $49.38 \pm 33.0$ | $64.71 \pm 37.8$ | 42.58 | 35.13 | 36.01 | 26.48 |
| Organ Average | $88.70 \pm 6.52$ | $96.48 \pm 4.03$ | 88.59 | 94.91 | 89.03 | 95.31 |

### 4.2   Qualitative results on validation set

Fig. 3 presents four representative segmentation results obtained from our final submission using the small nnU-Net. The top two rows depict case 17 and case 75, where the network successfully achieved high-accuracy identification of all organs. However, in the bottom two rows, specifically case 124 and case 94, noticeable segmentation deficiencies and over-segmentation errors are apparent. These issues may be attributed to the limited contextual information capturing capability of the small 3D nnU-Net and the loss of important details due to the large spacing of the resampled images, resulting in suboptimal segmentation of small-volume organs.

It is worth noting that compared to models trained solely on fully labeled data, the use of pseudo-labels in the semi-supervised approach improves segmentation accuracy. It can be observed that models employing pseudo-labels demonstrate higher generalization performance, highlighting the benefits of leveraging unlabeled data in these specific cases.

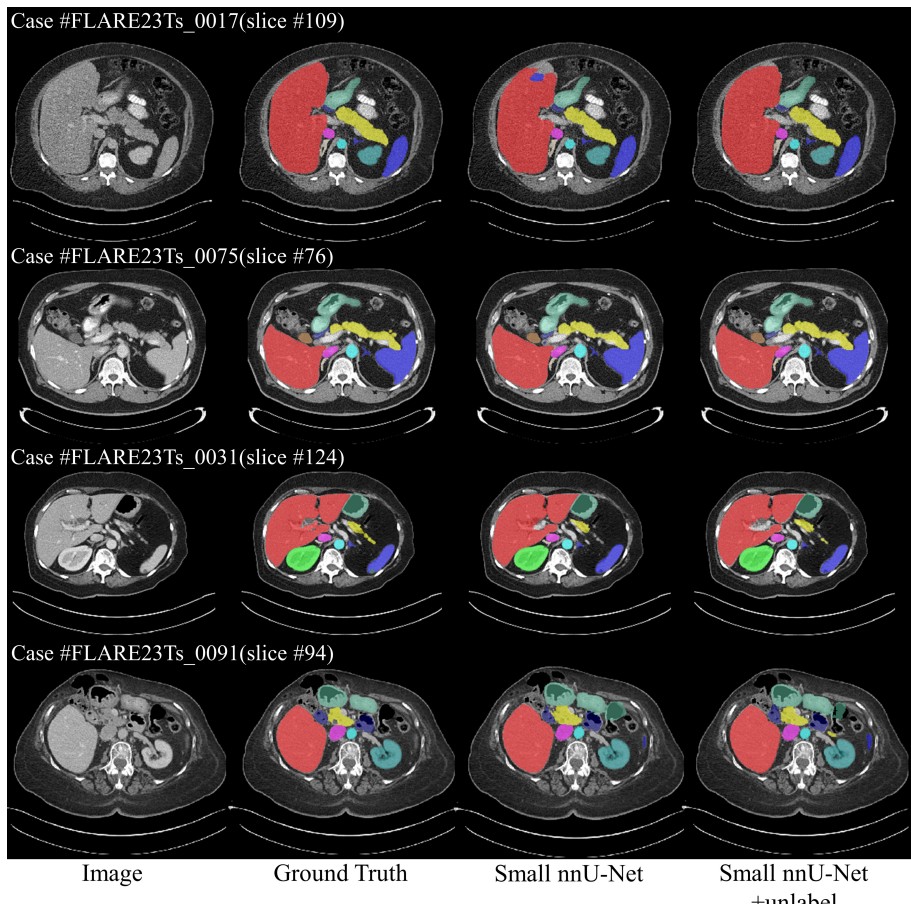

**Fig. 3.** Qualitative results of our small nnU-Net on two easy cases(case 17, case 75) and two challenging cases(case 124, case 94).

### 4.3 Segmentation Efficiency Results on Validation Set

Our validation Docker was submitted to an official evaluation platform with NVIDIA QUADRO RTX5000 (16G) and 28G RAM. The evaluation was performed on 100 validation sets, with an average running time of 29 seconds per

case. The average maximum GPU memory utilization was 3077MB, and the average area under the GPU memory time curve was 25411. We conducted an efficiency analysis by selecting validation set data of different sizes, and the results are presented in Table 5. Since our organ segmentation and tumor segmentation are separate models, we need to perform inference twice on the same sample, which contributes to longer run times. However, our GPU exhibits lower maximum memory usage.

**Table 5.** Quantitative evaluation of segmentation efficiency in terms of the running time and GPU memory consumption. (Total GPU denotes the area under GPU Memory-Time curve)

| Case ID | Image Size | Running Time (s) | Max GPU (MB) | Total GPU (MB) |
|---------|------------|------------------|--------------|----------------|
| 0001 | (512, 512, 55) | 27.73 | 2538 | 23063 |
| 0051 | (512, 512, 100) | 27.58 | 2538 | 25422 |
| 0017 | (512, 512, 150) | 31.46 | 3700 | 29963 |
| 0019 | (512, 512, 215) | 30.08 | 3700 | 25978 |
| 0099 | (512, 512, 334) | 34.74 | 1954 | 28417 |
| 0063 | (512, 512, 448) | 41.71 | 3700 | 34501 |
| 0048 | (512, 512, 499) | 46.18 | 3700 | 38808 |
| 0029 | (512, 512, 554) | 48.52 | 1690 | 39530 |

### 4.4   Effect of unlabeled data

In this study, we developed separate models for organ segmentation and tumor segmentation. Table 6 illustrates the impact of incorporating unlabeled data with labeled data during the training of the organ segmentation model. The results clearly demonstrate that this semi-supervised learning approach leads to enhanced performance compared to models trained solely on labeled data. It is worth noting that smaller models utilizing efficient inference strategies exhibit particularly significant performance improvements. Due to the time constraints of the competition, our tumor segmentation model was trained only on 597 fully labeled cases and did not utilize unlabeled data. Therefore, a comparative experiment between different training approaches for the tumor segmentation model could not be conducted.

**Table 6.** The segmentation accuracy of the default nnU-Net model and the small nnU-Net model in the validation set with or without the use of pseudo-labeled data.

| Method | Val Organ DSC(%) | Val Organ NSD(%) | Running Time |
|--------|------------------|------------------|--------------|
| Default nnU-Net | 87.51 | 92.92 | 4min |
| Default nnU-Net+unlabel | 89.40 | 94.31 | |
| Small nnU-Net | 85.49 | 90.94 | 29s |
| Small nnU-Net+unlabel | 88.59 | 94.91 | |

### 4.5   Limitation and future work

Several limitations have been discerned within the scope of our investigation. Due to time constraints during inference, we had to make compromises and increase the resample spacing, which has impacted the performance of model for small-scale targets. Exploring alternative resample strategies and optimizing this process could enhance the segmentation accuracy. Additionally, we have not investigated more complex neural network structures, such as the cascade 3D U-Net, which could potentially improve the ability of model to handle small targets. Future work should also include exploring advanced data augmentation techniques, such as CutMix, and employing sophisticated post-processing methods to further enhance the model's performance and robustness. Furthermore, the segmentation of organs and tumors is currently performed by two separate models, leading to low inference efficiency. Moving forward, our focus should be directed towards the integration of these two distinct models, with the overarching objective of attaining a unified framework proficient in undertaking both tasks. Addressing these limitations will contribute to refining and improving the applicability of our model.

## 5   Conclusion

This paper presents a semi-supervised learning framework based on nnU-Net, which leverages unlabeled data to enhance model performance and generalization. Additionally, we have made improvements to nnU-Net and compressed it to improve model inference efficiency while maintaining high accuracy to meet the requirements of the competition. Experimental results on the FLARE 2022 validation dataset demonstrate that, with an average inference time of 29 seconds for the lightweight nnU-Net, incorporating unlabeled data leads to an increase of 0.0310 in the DSC metric and 0.0397 in the NSD metric for the organ segmentation model, compared to models trained solely on labeled data. Ultimately, we have successfully developed a low-resource, fast-processing model for abdominal CT organ and tumor segmentation.

**Acknowledgements**  The authors of this paper declare that the segmentation method they implemented for participation in the FLARE 2023 challenge has not used any pre-trained models nor additional datasets other than those provided by the organizers. The proposed solution is fully automatic without any manual intervention. We thank all the data owners for making the CT scans publicly available and CodaLab [29] for hosting the challenge platform. This research was funded by the National Key Research and Development Program of China, grant number 2022YFF0710800; the National Key Research and Development Program of China, grant number 2022YFF0710802; the National Natural Science Foundation of China, grant number 62071311; the special program for key fields of colleges and universities in Guangdong Province (biomedicine and health) of China, grant number 2021ZDZX2008; and the Stable Support Plan for Colleges and Universities in Shenzhen of China, grant number SZWD2021010.

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

**Table 7.** Checklist Table. Please fill out this checklist table in the answer column.

| Requirements | Answer |
| --- | --- |
| A meaningful title | Yes |
| The number of authors ($\leq$6) | 6 |
| Author affiliations, Email, and ORCID | Yes |
| Corresponding author is marked | Yes |
| Validation scores are presented in the abstract | Yes |
| Introduction includes at least three parts: background, related work, and motivation | Yes |
| A pipeline/network figure is provided | Figure 1,2 |
| Pre-processing | Page 4 |
| Strategies to use the partial label | Page 4 |
| Strategies to use the unlabeled images. | Page 4,10 |
| Strategies to improve model inference | Page 5 |
| Post-processing | Page 5 |
| Dataset and evaluation metric section is presented | Page 4 |
| Environment setting table is provided | Table 1 |
| Training protocol table is provided | Table 2,3 |
| Ablation study | Page 10 |
| Efficiency evaluation results are provided | Table 5 |
| Visualized segmentation example is provided | Figure 3 |
| Limitation and future work are presented | Yes |
| Reference format is consistent. | Yes |