# OpenReview forum: "Semi-Supervised Abdominal Organ and Pan-cancer Segmentation with Efficient nnU-Net"
_MICCAI.org/2023/FLARE — Submitted to FLARE 2023_

### Official Review · Reviewer_edea · 2023-09-19
**Nice written paper but**

**Rating:** 7
**Confidence:** 5

**Review:**

## Summary:
The paper proposes an extension of nnU-Net for semi-supervised learning, focusing on abdominal organ and cancer segmentation. The authors improve the model's efficiency and ability to utilize unlabeled data through pseudo-labeling and student-teacher training. The results show strong performance on validation datasets.

## Strengths

1. The paper is well-written and presents a comprehensive view of the problem, methodology, and results. This makes it easy to follow and understand the contributions.

2. Significantly improves the inference speed of the original nnU-Net without compromising performance, making this a particularly valuable contribution.

## Weaknesses

1. The paper omits the description of spacing values used in the preprocessing steps, as spacing values can considerably impact both the model's accuracy and inference speed.

2. It would be highly beneficial if the authors could provide open-source code.

3. While time constraints are understood, it would be beneficial to later include results on the impact of unlabeled data for tumor segmentation in Table 6.

---

### Official Review · Reviewer_3KDZ · 2023-09-21
**Semi-Supervised Abdominal Organ and Pan-cancer Segmentation with Efficient nnU-Net**

**Rating:** 7
**Confidence:** 5

**Review:**

Summary

The author presents an exceptional approach to semi-supervised learning, which offers valuable insights for fellow scholars. However, in order to further enhance the quality of the paper, a few aspects can be improved upon.

Comments

1、The "Flare 21" in page 2 shoud be "FLARE21".

2、No tumor segmentation metrics were observed in the ablation experiment section for comparison.

---

### Official Review · Reviewer_xjv2 · 2023-09-21
**Semi-Supervised Abdominal Organ and Pan-cancer Segmentation with Efficient nnU-Net**

**Rating:** 7
**Confidence:** 5

**Review:**

Summary

This paper presents an extension of nnU-Net for semi-supervised learning. The authors improved the segmentation accuracy of the abdominal organ model by using pseudo-labeling of unlabeled data, while the effectiveness of the tumor segmentation model using unlabeled data remains to be explored. And the computational efficiency is improved with small size network and efficient inference.

Comments
1.  The article has some spelling errors, for example in the abstract "we reduce the network complexity of 3DU-Net", and "The top1 solution of Frare22 competition" in the introduction.
2. Runtime and GPU memory results should be shown within the abstract.
3. How to fuse organ and tumor model segmentation results is not described in the post-processing.

---

> ### Comment · Reviewer_xjv2 · 2023-11-20
> **2nd round Review**
>
> The new version of the paper has a good layout and is well structured. It is recommended for acceptance.

---

### Official Review · Reviewer_zkpy · 2023-10-02
**The paper proposes an extension to the nnU-Net framework by incorporating pseudo-label learning, achieving excellent results on the validation set. The approach effectively utilizes unlabeled data and improves inference efficiency.**

**Rating:** 7
**Confidence:** 5

**Review:**

Strengths:
1. Improved Utilization of Unlabeled Data: The proposed method generates high-quality pseudo-labels, enhancing segmentation accuracy.
2. Lightweight Student Model: The authors train a lightweight student model, improving inference efficiency and addressing resource constraints.
3. Impressive Results: The reported results on the validation set demonstrate the effectiveness of the proposed method.

Weakness:
1. Limited Innovation: The paper lacks significant innovation as it mainly focuses on integrating and optimizing pseudo-label learning with nnU-Net, and it is similar to the method [1].
2. Inadequate Comparison: The paper should include a comprehensive comparison with other state-of-the-art methods (especially, the method in [1]) to demonstrate the advantages of the proposed approach.

Reference:
[1] Huang Z, Wang H, Ye J, et al. Revisiting nnU-net for iterative pseudo labeling and efficient sliding window inference[M]. MICCAI Challenge on Fast and Low-Resource Semi-supervised Abdominal Organ Segmentation. Cham: Springer Nature Switzerland, 2022: 178-189.

---

### Decision · Program_Chairs · 2023-10-24

Accept